# Trends in inpatient and post-discharge mortality among young infants admitted to Kilifi County Hospital, Kenya: a retrospective cohort study

Alison Talbert [ID],[1] Moses Ngari,[1] Christina Obiero,[1,2] Amek Nyaguara,[1] Martha Mwangome,[1] Neema Mturi,[1] Nelson Ouma,[1] Mark Otiende,[1] James Berkley [ID] [1,3]

[1]KEMRI-Wellcome Trust Research Programme, Kilifi, Kenya
[2]Department of Global Health, University of Amsterdam, Amsterdam, The Netherlands
[3]Centre for Tropical Medicine & Global Health, University of Oxford, Oxford, UK

**Correspondence to**
Dr Alison Talbert;
ATalbert@kemri-wellcome.org

## ABSTRACT

**Objectives** To describe admission trends and estimate inpatient and post-discharge mortality and its associated exposures, among young infants (YI) admitted to a county hospital in Kenya.

**Design** Retrospective cohort study.

**Setting** Secondary level hospital.

**Participants** YI aged less than 60 days admitted to hospital from January 2009 to December 2019: 12 271 admissions in 11 877 individuals. YI who were resident within a Kilifi Health and Demographic Surveillance System (KHDSS): n=3625 with 4421 admissions were followed-up for 1 year after discharge.

**Primary and secondary outcome measures** Inpatient and 1-year post-discharge mortality, the latter in KHDSS residents.

**Results** Of 12 271 YI admissions, 4421 (36%) were KHDSS-resident. Neonatal sepsis, preterm complications and birth asphyxia accounted for 83% of the admissions. The proportion of YI among under-5s admissions increased from 19% in 2009 to 34% in 2019 (P_trend=0.02). Inpatient case fatality was 16%, with 66% of the deaths occurring within 48 hours of admission. The introduction of free maternity care in 2013 was not associated with a change in admissions or inpatient mortality among YI. During 1-year post-discharge, 208/3625 (5.7%) YI died, 64.3 (95% CI 56.2 to 73.7) per 1000 infant-years. 49% of the post-discharge deaths occurred within 1 month of discharge, and 49% of post-discharge deaths occurred at home. Both inpatient and post-discharge deaths were associated with low admission weight. Inpatient mortality was associated with clinical signs of disease severity, while post-discharge mortality was associated with the length of hospitalisation, leaving against advice and referral to a specialised hospital.

**Conclusions** YIs accounted for an increasing proportion of paediatric admissions and their overall mortality remains high. Post-discharge mortality accounts for a lower proportion of deaths but mortality rate is higher than among children aged 2–59 months. Services to address post-discharge mortality are needed and should focus on infants at higher risk.

## STRENGTHS AND LIMITATIONS OF THIS STUDY

⇒ Large sample size with systematic data collection.
⇒ Linkage of hospital admissions to a well-established demographic surveillance system, with low loss to follow-up.
⇒ Lack of accurate gestational age estimation or birth weight of most participants.
⇒ Data are from a single hospital and only the population covered by demographic surveillance.

## BACKGROUND

The United Nations Sustainable Development Goal 3 aims to ensure healthy living and promote well-being for all ages, with all countries aiming to reduce neonatal and under-5 mortality to below 12 and 25 per 1000 live births by 2030, respectively. In sub-Saharan Africa, child mortality has declined by ~58% in the last 30 years. However, the estimated neonatal and under-5 mortality rates in sub-Saharan Africa remained high in 2019 (27 and 76 per 1000 live births, respectively) with a similar neonatal mortality rate of 27 per 1000 live births in Kenya.[1] Combined neonatal and post-neonatal infant mortality accounts for over three-quarters of all under-5 deaths in Kenyan children.[2]

Young infants (YI) aged <60 days old comprise around half of the hospital admissions in sub-Saharan Africa and continue to face high risk of in-hospital mortality and long-term neurodisability.[3–6] Post-discharge mortality is emerging as a major problem in children in low-income and middle-income countries (LMICs),[7] however, there are limited data among YI. A systematic review of paediatric post-discharge mortality in developing countries included 24 studies published up to July 2017 with 19 from Africa.[8] Four studies included YI. Although young age was

BMJ

reported as a risk factor of mortality, no studies specifically identified deaths among infants aged <60 days. We have previously demonstrated excess post-discharge mortality among all hospitalised children, suggesting that hospitalisation itself selects vulnerable children with a sustained increased risk of dying over the longer-term.[7 9]

Better understanding of YI deaths occurring during hospitalisation and after discharge from hospital is vital for development and use of targeted interventions aimed at improving survival.

This analysis aimed to describe admission trends and measure inpatient and post-discharge mortality and its associated exposures, including the introduction of free maternity care, among YI admitted to Kilifi County Hospital (KCH), Kenya, and followed-up through the Kilifi Health and Demographic Surveillance System (KHDSS).

## METHODS
### Study participants and design
KCH is a secondary-level referral hospital situated in Kilifi County along the Kenyan coast. It serves a rural and peri-urban population. It has a maternity unit with approximately 6000 deliveries per year, a general paediatric ward with a newborn unit for babies aged less than 1 month and a paediatric High Dependency Unit (HDU) that also admits YIs. The year 2009 was selected as a starting point, because a previous analysis of mortality among YI covered admissions from 1990 to 2008.[10] Free maternity care was introduced by the Kenyan government on 1 June 2013 and led to a marked increase in health facility births.[11]

The KHDSS, established in 2002, covers a population of 279 158 within an area of 900 km$^2$ centred on KCH.[12] Census rounds visit each household every 4 months to ascertain vital status and migration in and out of the hospital catchment area.

We conducted a retrospective cohort study of YIs resident within the KHDSS who were admitted to KCH between 1 January 2009 and 31 December 2019. Children discharged alive and followed-up in KHDSS census rounds until March 2021 were eligible for analyses of post-discharge mortality. During the study period, there were nine health workers' strikes with the last nurses' strike lasting for 150 days (5 June 2017 to 2 November 2017)[13] (online supplemental table S1). For comparison, we also examined admissions aged 60 days to 59 months during the same period.

### Procedures
At admission, standardised medical history and clinical examination, including anthropometric measurements were obtained by trained clinical staff. Blood samples were systematically taken for complete blood count, slide for malaria microscopy and clinical chemistry, HIV antibody test and blood culture at hospital admission, as described previously.[14] A lumbar puncture for cerebrospinal fluid analysis was done at admission in infants in whom sepsis

was suspected and deferred in those seriously ill or with other contraindications. Clinical and laboratory data were recorded in real time on a ward surveillance database linked to the KHDSS database. Empiric antibiotics were initiated according to national guidelines[15] with ampicillin/benzylpenicillin plus gentamicin as first-line intravenous therapy. Second-line and subsequent antimicrobial therapy was guided by blood culture results and clinical progress. Mechanical ventilation was not available at KCH.

### Statistical methods
#### Study variables
Outcomes of interest were death in hospital and during 1 year after discharge. Exposures of interest were demographic, nutritional, clinical features and haematological, biochemical and microbiological findings at the time of admission. De-identified study data were deposited in the Harvard Dataverse depository.[16]

Weight at admission and mid-upper arm circumference (MUAC) were categorised as shown in table 1. Because approximately 40% of the YI were underweight (<2.5 kg), and 60% were aged ≤2 days at admission, YI's admission weights rather than anthropometric Z scores using WHO standards were reported. Furthermore, most YI who were born at home or in other hospitals and referred to KCH were missing gestational age estimates and birth weight to be able to estimate gestational age at birth using the INTERGROWTH 21st Newborn Size Standards (INSS).

Prematurity was defined as gestation age <37 weeks and low birth weight as birth weight <2500 g for YIs born at KCH. Admission blood glucose was categorised into <2.6, 2.6–7.0 and ≥7.0 mmol/L representing low, normal and high levels, respectively.[15] Missing data were not assumed to be missing at random. We, therefore, created categorical variables and added a missing category which was included in the regression analysis.

Demographic, anthropometric and clinical data are presented as frequencies and proportions for categorical variables and means (SD) or median (IQR) for continuous variables depending on the underlying distribution. Proportions of missing data for each variable are shown in online supplemental table S2.

Monthly admissions and case fatality were plotted against time (month of admission) to visually inspect the trend from 2009 to 2019 and the predicted trend line superimposed on the curves. We used the Augmented Dickey Fuller test to test if the time series were stationary (no trend or seasonal effects). We also presented annual absolute admissions, proportion of YI among all admissions <60 months and case fatality. Monthly admissions and case fatality were tested for annual linear trend using an extension of the Wilcoxon rank-sum test of trend across ordered groups.[17]

We used interrupted time series analysis to estimate the level and trend changes before and after introduction of free maternity care (1 June 2013). We created a time month variable coded sequentially from January 2009 to

**Table 1** Study participants characteristics at admission

| | All young infant admissions (N=12271)* | Young infant admissions KHDSS residents (N=4421) | Young infant admissions non-KHDSS residents (N=7850) | P value |
|---|---|---|---|---|
| Demographics | | | | |
| Age in days | | | | |
| 0–2 | 7856 (64) | 2731 (62) | 5125 (65) | <0.001 |
| 3–7 | 1384 (11) | 468 (11) | 916 (12) | |
| 8–28 | 1506 (12) | 587 (13) | 919 (12) | |
| >28 | 1525 (12) | 635 (14) | 890 (11) | |
| Sex (female) | 5245 (43) | 1900 (43) | 3345 (43) | 0.70 |
| Reported born premature | 2970 (24) | 1019 (23) | 1951 (25) | 0.005 |
| Reported low birth weight | 1782 (15) | 581 (13) | 1201 (15) | <0.001 |
| Born at KCH | | | | |
| Yes | 6757 (55) | 2743 (62) | 4014 (51) | <0.001 |
| No | 5514 (45) | 1678 (38) | 3836 (49) | |
| Anthropometry | | | | |
| Weight (kg) | | | | |
| <1.5 | 1767 (14) | 566 (13) | 1201 (15) | <0.001 |
| 1.5 to <2.5 | 3211 (26) | 1128 (26) | 2083 (27) | |
| ≥2.5 | 7193 (59) | 2684 (61) | 4509 (57) | |
| Missing | 100 (0.8) | 43 (1.0) | 57 (0.7) | |
| MUAC (cm) | | | | |
| <9 | 3933 (32) | 1342 (30) | 2591 (33) | <0.001 |
| 9–10 | 2492 (20) | 862 (20) | 1630 (21) | |
| 10–11 | 2926 (24) | 1035 (23) | 1891 (24) | |
| ≥11 | 2622 (21) | 1056 (24) | 1566 (20) | |
| Missing | 298 (2.4) | 126 (2.9) | 172 (2.2) | |
| Clinical features | | | | |
| Axillary temperature | | | | <0.001 |
| <36°C | 3553 (29) | 1358 (31) | 2195 (28) | |
| 36–37.5°C | 4692 (38) | 1711 (39) | 2981 (38) | |
| >37.5°C | 3948 (32) | 1318 (30) | 2630 (34) | |
| Respiratory rate/min† | | | | |
| Bradypnoea | 540 (4.4) | 188 (4.3) | 352 (4.5) | 0.56 |
| Normal | 7333 (60) | 2647 (60) | 4686 (60) | |
| Tachypnoea | 4158 (34) | 1490 (34) | 2668 (34) | |
| Missing | 240 (2.0) | 96 (2.2) | 144 (1.8) | |
| Heart rate/min‡ | | | | |
| Bradycardia | 396 (3.2) | 158 (3.6) | 238 (3.0) | 0.11 |
| Normal | 8162 (67) | 2910 (66) | 5252 (67) | |
| Tachycardia | 3667 (30) | 1331 (30) | 2336 (30) | |
| Missing | 46 (0.4) | 22 (0.5) | 24 (0.3) | |
| Hypoxia§ | 2668 (22) | 932 (21) | 1736 (22) | 0.19 |
| Lower chest wall indrawing | 5562 (45) | 2051 (46) | 3511 (45) | 0.13 |
| Wheeze | 112 (0.9) | 46 (1.0) | 66 (0.8) | 0.41 |
| Stridor | 62 (0.5) | 19 (0.4) | 43 (0.6) | 0.48 |

**Table 1** Continued

| | All young infant admissions (N=12 271)* | Young infant admissions KHDSS residents (N=4421) | Young infant admissions non-KHDSS residents (N=7850) | P value |
|---|---|---|---|---|
| Breathing difficulty | 5966 (49) | 2172 (49) | 3794 (48) | 0.44 |
| Cyanosis | 560 (4.6) | 210 (4.8) | 350 (4.5) | 0.54 |
| Capillary refill >2 s | 301 (2.6) | 105 (2.4) | 196 (2.5) | 0.81 |
| Temperature gradient | 710 (5.8) | 258 (5.8) | 452 (5.8) | 0.73 |
| Weak pulse | 463 (3.8) | 157 (3.6) | 306 (3.9) | 0.05 |
| Lethargy | 971 (7.9) | 325 (7.4) | 646 (8.2) | 0.15 |
| Impaired consciousness¶ | 792 (6.5) | 250 (5.7) | 542 (6.9) | 0.007 |
| Bulging fontanel | 111 (0.9) | 32 (0.7) | 79 (1.0) | 0.21 |
| Stiff neck | 48 (0.4) | 10 (0.2) | 38 (0.5) | 0.05 |
| Convulsions | 689 (5.6) | 197 (4.5) | 492 (6.3) | <0.001 |
| Sunken eyes | 134 (1.1) | 44 (1.0) | 90 (1.2) | 0.44 |
| Reduced skin turgor | 308 (2.5) | 97 (2.2) | 211 (2.7) | 0.19 |
| Pallor | 633 (5.2) | 221 (5.0) | 412 (5.3) | 0.55 |
| Laboratory features | | | | |
| Meningitis** | 98 (0.8) | 33 (0.8) | 65 (0.8) | 0.87 |
| Haemoglobin (<110 g/L)†† | 1207 (9.8) | 476 (11) | 731 (9.3) | 0.02 |
| HIV antibody positive | 441 (3.6) | 142 (3.2) | 299 (3.8) | 0.11 |
| Malaria slide positive | 5 (0.04) | 4 (0.09) | 1 (0.01) | 0.02 |
| Bacteraemia | 590 (4.8) | 170 (3.9) | 420 (5.4) | <0.001 |
| White blood cells ($10^9$cells/L)‡‡ | | | | |
| <4 | 134 (1.1) | 54 (1.2) | 80 (1.0) | <0.001 |
| 4–20 | 8738 (71) | 3228 (73) | 5510 (70) | |
| >20 | 2202 (18) | 690 (16) | 1512 (19) | |
| Unavailable | 1197 (9.8) | 449 (10) | 748 (9.5) | |
| Platelets ($10^9$/L)§§ | | | | |
| <150 | 1615 (13) | 586 (13) | 1029 (13) | 0.59 |
| ≥150 | 9455 (77) | 3387 (77) | 6068 (77) | |
| Unavailable | 1201 (9.8) | 448 (10) | 753 (9.6) | |
| Blood glucose (mmols/L) | | | | |
| <2.6 | 2479 (20) | 882 (20) | 1597 (20) | 0.29 |
| 2.6–7.0 | 5086 (41) | 1875 (42) | 3211 (41) | |
| >7.0 | 688 (5.6) | 231 (5.2) | 457 (5.8) | |
| Unavailable | 4018 (33) | 1433 (32) | 2585 (33) | |

*Eligible admissions were young infants aged <60 days admitted from 2009 to 2019.
†Tachypnoea: respiratory rate ≥60 breaths/min, bradypnoea: respiratory rate <30 breaths/min.
‡Tachycardia: heart rate >160 beats/min, bradycardia: heart rate <100 beats/min.
§Hypoxia: oxygen saturation <90%.
¶Impaired consciousness level if 'prostrate' or 'unconscious'.
**Meningitis: positive CSF culture, or positive CSF microscopy, or positive CSF antigen test, or elevated CSF WBC count (≥0.02x $10^9$ cells/Lin young infants aged 0–28 days OR, ≥0.01x$10^9$ cells/L in young infants aged 29–59 days) PLUS a positive blood culture for a known pathogen.
††Anaemia: haemoglobin <110g/L.
‡‡Normal values WBC 4–20×$10^9$ cells/L, leucopenia WBC <4×$10^9$cells/L, leucocytosis WBC >20×$10^9$ cells/L.
§§Normal values platelets ≥150×$10^9$ /L, thrombocytopenia <150×$10^9$/L.
CSF, cerebrospinal fluid ; KCH, Kilifi County Hospital; KHDSS, Kilifi Health and Demographic Surveillance System; MUAC, mid-upper arm circumference; WBC, white blood cell.

December 2019 and a binary variable coded as 0 and 1 for admissions before and after June 2013, respectively, to represent introduction of free maternity care. We defined seasonal effect variables using month of the year modelled on harmonic terms using the Fourier code in Stata. To measure the effect of free maternity care, we used the negative binomial regression model because of the presence of overdispersion in the trends and reported regression coefficients transformed into incidence rate ratios (IRR). All the negative binomial regression models included the following independent variables: the time month variable, the binary pre and post free maternity care variable and the seasonal effect variable.

Since YIs could be admitted more than once while <60 days old, we included multiple admissions using unique IDs and adjusted for clustering by individual with robust SEs. To identify exposures associated with inpatient death, we treated being discharged alive as a competing event and fitted the proportional subdistribution hazard model using the Fine-Gray competing risk model.[18] The measure of effect reported from the model was the subdistribution HRs (SHR) and their respective 95% CIs. To build the multivariable regression model, a backward stepwise approach was used where all the independent variables assessed in the univariate models were included in the model and only those with a p value<0.1 retained in the final multivariable model.

For the post-discharge analysis, only data from those YI discharged alive and resident within the KHDSS were analysed. Time at risk was defined from the date of discharge to 365 days later or censure at the date of death or outmigration from the KHDSS. We performed a 'multiple discharges' analysis where YI with multiple admissions had their follow-up time reset at each successive discharge date. Exposures associated with post-discharge were assessed using a gamma distribution shared frailty Cox proportional hazards regression model accounting for YI with multiple discharges. The proportional hazards assumption was assessed using the scaled Schoenfeld residuals test (online supplemental tables S3 and S4). All exposures assessed in the univariate models were considered for inclusion in the multivariable Cox proportional hazards regression model using a backward stepwise approach similar to the inpatient analysis. Both the inpatient and post-discharge multivariable regression models' discrimination performance were assessed using bootstrapped area under receiver operating characteristic curves (AUC) replicated 1000 times.

As sensitivity analysis, we assessed the YI born at KCH and enrolled to the Kilifi Perinatal and Maternal Research Project, which had collected comprehensive birth data including birth weight and gestational age (weeks).[19] We estimated their birth weight Z scores using the INSS and ran the regression models replacing admission weight with birth weight Z score.[20]

Statistical significance was evaluated using 95% CI and a two-tailed p value<0.05. Statistical analyses were

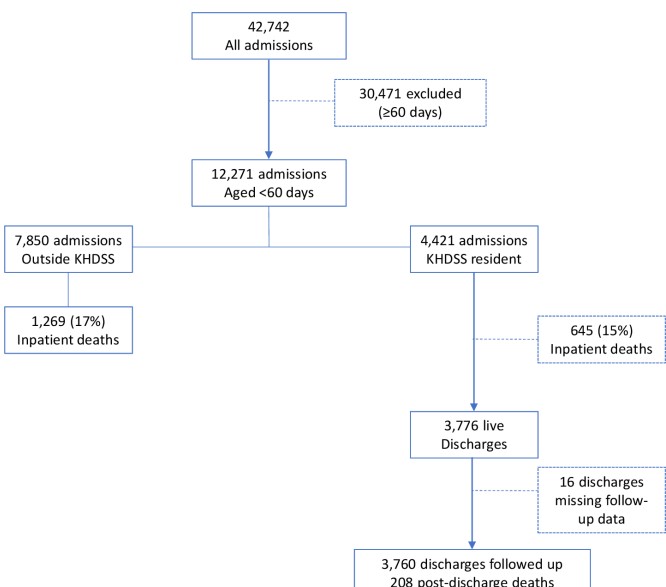

**Figure 1** Flow of study participants. KHDSS, Kilifi Health and Demographic Surveillance System.

conducted using Stata V.17.0 (College Station, Texas, USA).

## Study size
We used all available eligible YI data from 2009 to 2019 (4421 for inpatient and 3625 for post-discharge analyses) regardless of sample size.

## Ethical considerations
Written consent was provided by the caregivers of all the surveillance study participants.

## Patient and public involvement
There was no patient and public involvement in the planning or execution of this retrospective cohort study.

# RESULTS
## Baseline characteristics
During the study period, there were 42 742 paediatric admissions to KCH, of which 12 271 (29%) admission events among 11 877 individuals were aged <60 days. Of the 12 271 YI admission events, 4421 (36%) were resident in the KHDSS and included in the analysis (figure 1). This comprised 4272 individual YI: 4131 with one admission, 133 with two admissions and 8 with three admissions within the first 60 days of life.

## KHDSS-resident admissions
Among the 4421 YI admission events among KHDSS residents, 2731 (62%) were ≤2 days old and 1900 (43%) were women. Reported prematurity and low birth weight were 1019 (23%) and 581 (13%), respectively. Low weight (<2.5 kg) was observed in 1694 YIs (38%) while 1342 (30%) had MUAC <9.0 cm. Common presenting clinical signs were lower chest wall indrawing (46%) and breathing difficulty (49%). Thirty per cent had fever, 31%

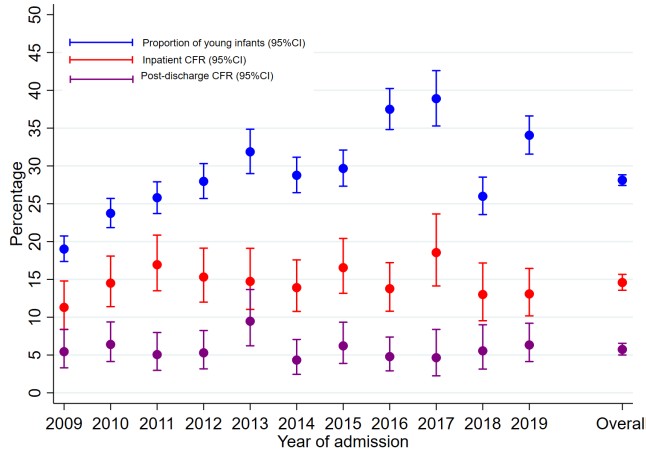

**Figure 2** Annual proportion of young infants admissions to all admissions <60 months, inpatient case fatality ratio (CFR) and post-discharge CFR. Proportions are plotted with 95% CIs.

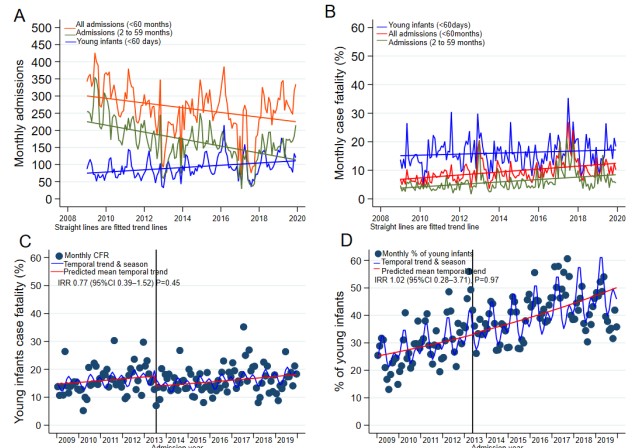

**Figure 3** (A) Monthly hospital admissions (with predicted mean temporal trend), (B) monthly case fatality rates (with predicted mean temporal trend), (C) monthly young infant inpatient case fatality before and after June 2013 and (D) monthly proportions of young infants to admissions <60 months old. CFR, case fatality ratio; IRR, incidence rate ratio.

had hypothermia and 30% tachycardia. Nine hundred and thirty-two YI (21%) had hypoxia (SaO2 <90%) at admission and 250 (5.7%) had impaired consciousness. Presenting signs at admission for all the YI stratified by KHDSS residence are shown in table 1. Malaria was rare (n=4, 0.09%) while 142 (3.2%) and 170 (3.9%) YI were HIV antibody positive and had bacteraemia, respectively. Online supplemental table S3 lists the bacterial isolates that were presumed pathogens, led by *Klebsiella pneumoniae*, *Escherichia coli*, *Staphylococcus aureus* and Group B Streptococcus.

### Admissions over time

The annual number of admissions are shown in online supplemental table S4. The overall proportion of YI among all admissions under-5 years old was 28% (95% CI 27% to 29%), increasing from 19% in 2009 to 34% in 2019 (test of linear trend p=0.02) (figure 2). Figure 3A shows the upward trend of absolute YI admissions and downward trends for 2–59 months old and all admissions <60 months old (all p values for tests for stationarity <0.05). There was no significant difference in monthly YI admissions before introduction of free maternity care in June 2013 (monthly median (IQR) of 76 (66–96) admissions) and after June 2013 (monthly median (IQR) of 95 (78–125) admissions) season-adjusted IRR 1.06 (95% CI 0.54 to 2.09) p=0.86 (online supplemental figure S1A). The mean monthly YI admissions on day of birth did not differ before and after June 2013; season-adjusted IRR 0.88 (95% CI 0.44 to 1.76), p=0.72. The proportion of YI admissions to total admissions aged <60 months before and after June 2013 were not different; season-adjusted IRR 1.02 (95% CI 0.28 to 3.71) p=0.97 (figure 3D). We found no significant difference in monthly absolute admissions (all admissions <60 months old), before and after June 2013; season-adjusted IRR 1.01 (95% CI 0.51 to 2.00) p=0.97 (online supplemental figure S1B).

### Inpatient deaths

Overall, 1914/11 877 (16%) of YI died in hospital. The risk of inpatient death was not significantly different between 645/4272 (15%) KHDSS residents and 1269/7605 (17%) non-residents of KHDSS (age-adjusted and sex-adjusted SHR 0.93 (95% CI 0.85 to 1.02) p=0.12) (figure 1). The annual YI inpatient case fatality ratio was stable (11% in 2009 and 13% in 2019. P value for trend=0.80) (figure 2). Monthly inpatient case fatality for YI, 2–59 months old and all <60 months old children are shown in figure 3B.

During the study period there were 3119 inpatient deaths among admissions <60 months old admitted at KCH, with YI admissions accounting for 61% (95% CI 60% to 63%) of the deaths and no significant linear trend from 2009 to 2019 (trend p=0.29). The mean monthly YI inpatient case fatality was 16% (SD 0.86) and 16% (SD 1.23) before and after June 2013, respectively; season-adjusted IRR 0.77 (95% CI 0.39 to 1.52) p=0.45 (figure 3C). The mean monthly case fatality for all admissions aged <60 months and admissions 2–59 months old did not differ before June 2013 and after June 2013; season-adjusted IRR 0.79 (95% CI 0.39 to 1.58) p=0.50 and IRR 0.81 (95% CI 0.39 to 1.69) p=0.57, respectively (online supplemental figure S1C,D).

Among the 4421 KHDSS-resident YI admissions, median (IQR) time to death was 2 (1–4) days, while the survivors were admitted for 5 (3–8) days. A total of 423/645 (66%) deaths occurred within the first 48 hours following admission. Forty-one YI left against medical advice, and 55 were referred to other hospitals for further care.

### Admission diagnosis and case fatality ratio

The most common reasons for hospital admission were neonatal sepsis (47%), preterm complications (20%) and birth asphyxia (16%) accounting for 83% of all YI admissions (table 2). The case fatality ratios for YI with

**Table 2** Discharge diagnosis assigned by clinician

| Discharge diagnosis* | No. (%) diagnosis assigned by clinician at discharge | |
| --- | --- | --- |
| | All admissions (N=4421) | Inpatient deaths (N=645) |
| Neonatal sepsis | 2097 (47) | 201 (9.6) |
| Preterm complications | 889 (20) | 262 (29) |
| Birth asphyxia | 724 (16) | 201 (28) |
| Neonatal jaundice | 611 (14) | 56 (9.2) |
| Lower respiratory tract infection | 486 (11) | 41 (8.4) |
| Respiratory distress syndrome | 263 (6.0) | 136 (52) |
| Congenital anomalies | 215 (4.9) | 55 (26) |
| Meningitis† | 112 (2.5) | 11 (9.8) |
| Anaemia | 78 (1.8) | 14 (18) |
| Malnutrition | 36 (0.8) | 1 (2.8) |
| None specified | 69 (1.6) | 4 (5.8) |
| Others | 266 (6.0)‡ | 13 (4.9) |

*Young infant could be assigned up to two diagnoses.
†Meningitis: positive CSF culture, or positive CSF microscopy, or positive CSF antigen test, or elevated CSF WBC count (≥0.02x10⁹/L in young infants aged 0–28 days OR, ≥0.01x10⁹/L in young infants aged 29–59 days) PLUS a positive blood culture for a known pathogen.
‡Accidents-3, acute abdominal obstruction-15, bronchiolitis-12, burns-1, candidiasis-1, cellulitis abscess-21, chickenpox-1, chromosomal abnormality-5, cerebral abscess-1, conjunctivitis-2, dehydration-2, dental problems-1, diabetes-1, elective surgery-5, encephalopathy-9, epilepsy-7, extra pulmonary tuberculosis-1, febrile convulsions-5, feeding difficulty-1, gastroenteritis-15, haemolytic uraemic syndrome-1, hydrocephalus-11, laryngotracheobronchitis/croup-1, immunosuppression-17, malaria-2, male genital problem-1, meconium aspiration-33, neonatal haemorrhage-14, neonatal tetanus-10, other skin disease-3, otitis media-1, poisoning (organophosphates)–1, pyogenic arthritis-1, rabies-1, rash-4, renal failure-6, trauma/fractures/road traffic accident-11, urinary tract infection-10, upper respiratory tract infection-24, viral hepatitis-2, viral infection-3.
CSF, cerebrospinal fluid; WBC, white blood cell.

respiratory distress syndrome, preterm complications and birth asphyxia were 52%, 29% and 28%, respectively (table 2).

### Exposures associated with inpatient death
Variables assessed for association with inpatient death in univariate models are shown in online supplemental table S5. In the multivariable analysis (table 3), admissions at age ≤2 days and 3–7 days, compared with ≥28 days old, were associated with inpatient deaths. Being born at KCH was not associated with inpatient death, so was not included in the multivariate analysis. Very low admission weight (<1.5 kg) and weight 1.5–2.4 kg compared with ≥2.5 kg were positively associated with inpatient deaths. Signs of clinical severity (bradypnoea, tachypnoea, bradycardia, hypoxia, lower chest wall indrawing, breathing difficulty, weak pulse, impaired consciousness and hypothermia, but not fever), meningitis, bacteraemia, leucopenia and leucocytosis but not an HIV antibody positive test (adjusted SHR 1.15 (95% CI 0.81 to 1.63)) were positively associated with inpatient death. The multivariable model bootstrapped AUC was 0.88 (95% CI 0.86 to 0.89) (table 3). Performance of a multivariable model including only 4272 single admissions did not differ from the model with multiple admissions (bootstrapped AUC 0.88 (95% CI 0.86 to 0.89)) (online supplemental table S6).

### Post-discharge death
There were 3776 live discharges from 3640 YI residents of KHDSS, of which 3760 (from 3625 individual YI) were followed-up for 3233 infant-years (figure 1). During 1-year follow-up, there were 208/3625 (5.7%) deaths: 64.3 (95% CI 56.2 to 73.7) deaths per 1000 infant-years. The median (IQR) time to death after discharge was 35 (7–92) days. Of the 208 post-discharge deaths, 101 (49%), 160 (77%), 179 (86%) and 193 (93%) occurred within 1, 3, 6 and 9 months after discharge, respectively. The annual YI post-discharge case fatality was 5.4% in 2009 and 6.3% in 2019 without evidence of linear trend (p value for trend=0.77) (figure 2).

One hundred and one (49%) of the 208 post-discharge deaths occurred at home without hospital readmission, 67 (32%) occurred during readmission to KCH and 40 (19%) occurred at other health facilities. The five leading assigned causes of deaths for the 67 deaths at KCH were: neonatal sepsis (24%), preterm complications (22%), congenital heart disease (15%), neonatal jaundice (7.4%) and meningitis (7.4%) which were similar to index admission diagnosis (online supplemental table S7). Causes of other deaths were unknown.

Overall, we observed 853 (20%) deaths among 4272 individual YIs: 645 inpatient and 208 post-discharge, hence 24% of deaths were post-discharge.

Exposures assessed for association with post-discharge mortality are shown in online supplemental table S8. In the multivariable Cox regression model, born outside KCH, log days of hospital admission, leaving against advice and referral to more specialised hospital were positively associated with post-discharge mortality. Other exposures associated with post-discharge mortality were low admission weight, fever and weak pulse (table 3). The multivariable model bootstrapped AUC was 0.76 (95% CI 0.73 to 0.80).

### Subgroup analysis
In a subgroup analysis including 1358 admissions of YIs born at KCH, their median (IQR) gestational age was 38 (36–40) weeks and birth weight 2778 (2000–3195) g, respectively. In the univariate regression model, born premature, low birth weight and birth weight Z score <−2 were positively associated with inpatient mortality (online supplemental table S9). In the multivariable model, low birth weight, admission age <8 days, bacteraemia and

**Table 3** Multivariable regression analysis of factors associated with inpatient and post-discharge mortality

| | Inpatient analysis | | Post-discharge analysis | |
|---|---|---|---|---|
| | Adjusted SHR* | P value | Adjusted HR | P value |
| Demographics | | | | |
| Age in days | | | | |
| 0–2 | 2.12 (1.46–3.06) | <0.001 | 1.30 (0.73–2.31) | 0.37 |
| 3–7 | 3.88 (2.46–6.10) | <0.001 | 0.80 (0.38–1.68) | 0.56 |
| 8–28 | 1.42 (0.90–2.25) | 0.13 | 1.45 (0.81–2.59) | 0.21 |
| >28 | Reference | | Reference | |
| Sex (female) | 0.91 (0.78–1.07) | 0.26 | 0.98 (0.74–1.31) | 0.94 |
| Born at KCH | | | | |
| Yes | † | | Reference | |
| No | † | | 1.59 (1.18–2.14) | 0.003 |
| Admission days (log) | † | | 1.87 (1.54–2.26) | <0.001 |
| Type of discharge | | | | |
| Normal | † | | Reference | |
| Absconded | † | | 3.01 (1.22–7.46) | 0.02 |
| Transferred/referred | † | | 12.8 (8.11–20.2) | <0.001 |
| Anthropometry | | | | |
| Weight (kg) | | | | |
| <1.5 | 2.16 (1.75–2.67) | <0.001 | 1.95 (1.38–2.74) | <0.001 |
| 1.5 to <2.5 | 1.42 (1.16–1.74) | 0.001 | 0.82 (0.48–1.42) | 0.49 |
| ≥2.5 | Reference | | Reference | |
| Missing weight | 3.85 (2.59–5.71) | <0.001 | – | |
| Clinical features | | | | |
| Axillary temperature | | | | |
| <36°C | 1.44 (1.17–1.78) | 0.001 | 1.06 (0.74–1.53) | 0.75 |
| 36–37.5°C | Reference | | Reference | |
| >37.5°C | 1.09 (0.84–1.41) | 0.53 | 0.69 (0.47–0.99) | 0.04 |
| Missing temperature | 1.03 (0.38–2.75) | 0.96 | 1.09 (0.15–8.22) | 0.93 |
| Respiratory rate/min | | | | |
| Bradypnoea | 1.45 (1.09–1.93) | 0.01 | 1.66 (0.76–3.63) | 0.21 |
| Normal | Reference | | Reference | |
| Tachypnoea | 0.80 (0.67–0.95) | 0.01 | 1.24 (0.93–1.66) | 0.14 |
| Missing | 1.51 (0.64–3.56) | 0.34 | 0.80 (0.11–5.82) | 0.82 |
| Heart rate/min | | | | |
| Bradycardia | 1.40 (1.08–1.82) | 0.01 | † | |
| Normal | Reference | | | |
| Tachycardia | 1.14 (0.94–1.37) | 0.18 | † | |
| Missing | 0.41 (0.03–5.13) | 0.49 | † | |
| Hypoxia (SaO2 <90%) | 1.62 (1.37–1.92) | <0.001 | † | |
| Capillary refill >2 s | 1.34 (0.97–1.86) | 0.08 | † | |
| Lower chest wall indrawing | 1.41 (1.14–1.75) | 0.002 | † | |
| Stridor | 1.93 (0.92–4.03) | 0.08 | † | |
| Breathing difficulty | 1.45 (1.15–1.82) | 0.001 | † | |
| Weak pulse | 1.61 (1.19–2.17) | 0.002 | 2.22 (1.01–4.89) | 0.04 |
| Bulging fontanel | 2.45 (0.91–6.65) | 0.08 | 2.59 (0.92–7.26) | 0.07 |

Continued

**Table 3** Continued

| | Inpatient analysis | | Post-discharge analysis | |
|---|---|---|---|---|
| | Adjusted SHR* | P value | Adjusted HR | P value |
| Impaired consciousness | 2.21 (1.72–2.84) | <0.001 | † | |
| Pallor | 1.30 (0.98–1.71) | 0.07 | † | |
| Laboratory features | | | | |
| Meningitis | 5.45 (2.50–11.8) | <0.001 | 2.16 (0.73–6.37) | 0.17 |
| HIV antibody positive | 1.15 (0.81–1.63) | 0.43 | 0.94 (0.43–2.05) | 0.87 |
| Bacteraemia | 2.21 (1.51–3.22) | <0.001 | † | |
| White blood cells ($10^9$ cells/L) | | | | |
| <4 | 2.17 (1.30–3.62) | 0.003 | † | |
| 4–20 | Reference | | † | |
| >20 | 1.71 (1.43–2.04) | <0.001 | † | |
| Unavailable | 1.09 (0.82–1.44) | 0.57 | † | |
| Model performance | | | | |
| Bootstrapped AUC (95% CI) | 0.88 (0.86 to 0.89) | | 0.76 (0.73 to 0.80) | |

*The SHR are from the Fine and Gray's proportional subhazards model, HR from the shared frailty Cox regression model.
†Variables not selected for inclusion in the multivariable model, AUC; area under receiver operating characteristics. Meningitis: positive CSF culture, or positive CSF microscopy, or positive CSF antigen test, or elevated CSF WBC count (≥0.02x10⁹/L in young infants aged 0–28 days OR, ≥0.01 x 10⁹/L in young infants aged 29–59 days) PLUS a positive blood culture for a known pathogen.
CSF, cerebrospinal fluid ; KCH, Kilifi County Hospital; SHR, subdistribution HR; WBC, white blood cell.

signs of clinical severity were associated with inpatient mortality (online supplemental table S10).

Among the 1142 YI followed-up for 1021 child/years of which 41/1142 (3.6%) died, low birth weight (adjusted HR 2.76 (95% CI 1.30 to 5.82)) was positively associated with post-discharge mortality in the multivariable model (online supplemental table S10).

## DISCUSSION
### Trends in admissions and proportions of YI
During the study period, we observed a marked increase in YI admissions and the proportion of YI among admissions in under-5s increased from around one-fifth in 2009, to more than one-third in 2019. However, this did not seem to be associated with the introduction of free maternity care in 2013. Lack of observable effect may be due to challenges faced during policy implementation arising from inadequate expansion of quality healthcare facilities and resources. Several authors reported an increase in mothers attending Kenyan health facilities for antenatal care and delivery,[11 21] however our results suggest this occurred in the context of a general trend which we previously observed during 1990–2008.[10]

Conversely, the number of admitted children older than 60 days decreased alongside a reduction in local malaria transmission,[22] introduction of routine childhood pneumococcal conjugate and rotavirus immunisation[23] and expansion in numbers of health facilities in Kilifi County.[24] Variation in annual admissions over the years was due to multiple health workers' strikes.[13] During these periods, the general paediatric ward was closed and

only the sickest children were admitted to the paediatric HDU due to limited staffing and bed capacity. The time series analysis indicated an increase in inpatient mortality during strikes (figure 3C).

The leading diagnoses at admission in our analysis were neonatal sepsis, preterm complications and birth asphyxia, similar to the period 1990–2008.[10] Over one-third of the admissions from KCH maternity were preterm and the hospital also received referrals of preterm and very low birthweight infants from subcounty hospitals and local health centres. There are few African published data sets of neonatal or YI inpatient diagnoses; in a network of seven Nigerian and Kenyan hospitals, prematurity accounted for over half (52%) and birth asphyxia almost a quarter (24%) of neonatal admissions.[25] The leading bacterial isolates from blood cultures in our study (*K. pneumoniae*, *E. coli*, *S. aureus*) were similar to those among YI in rural settings of Tanzania and Burkina Faso.[26] Kenya attained elimination status of maternal and neonatal tetanus in 2018, following immunisation campaigns in high-risk regions.[27] Compared with 1990–2008,[10] neonatal tetanus was uncommon at our centre with only 10 cases in 11 years.

### Inpatient deaths
The WHO has reported that in 2019, '47% of all under-5 deaths occurred in the newborn period with about one-third dying on the day of birth and close to three-quarters dying within the first week of life'.[28] Delivery by a skilled health worker has been shown to be effective in reducing perinatal mortality.[29] We did not collect data on delivery by a skilled birth attendant but in 2018/2019 69% of the

births in Kilifi County were reported to be attended by skilled health personnel which is slightly higher than the national average.[30] We found YI accounted for more than 60% of under-5s inpatient deaths, similar to a retrospective study of 16 Kenyan public hospitals in which neonatal deaths comprised 66% of the inpatient paediatric deaths.[5] We found respiratory distress syndrome, birth asphyxia and preterm complications had the highest inpatient mortality. Mechanical ventilation was not available in KCH. Improvements in peripartum care of mothers and infants together with appropriate technology such as non-invasive ventilation for management of respiratory complications of preterm birth are priorities for reduction in neonatal mortality in hospitals in LMICs.[5]

### Post-discharge deaths

Less than a quarter (24%) of all deaths during 1 year of follow-up occurred post-discharge. This reflects a high inpatient (16%) case fatality rate with many very early inpatient deaths compared with 6.6% in children aged ≥60 days.[7] Nevertheless, the post-discharge YI mortality rate (64.3 per 1000 child/years) was more than twice that of a cohort of children aged 2–59 months admitted to KCH between 2007 and 2015.[31] This reflects post-discharge mortality rates being highest in younger age groups, such as in Tanzania among under 1 year olds: 72 per 1000 child/years (95% CI 67.2 to 77.2) falling to 6.9 (95% CI 5.5 to 8.7) per 1000 child/years in 4 to <5 years old.[32]

A greater proportion of YI post-discharge deaths occurred in hospital than among older children,[7] implying that caregivers may be more likely to seek readmission for YI or may live closer to KCH. About half of the post-discharge deaths occurred within the first month, highlighting the need for formal 'down-referral' for continuity of care after discharge in high-risk YI.

Analysis of exposures revealed that some were common for both inpatient and post-discharge mortality: low admission weight, axillary temperature and respiratory rate. Birth weight was not available for most YI but low admission weight <2.5 kg was common (40%) in our participants. In YI it is difficult to distinguish low birth weight from malnutrition, but we have reported the higher case fatality rates in the lower admission weight categories (online supplemental tables S3 and S4). Of known causes of post-discharge deaths, leading ones were related to problems in the early neonatal period.

### Strengths and limitations of the study

Strengths of this study are large sample size, systematic collection of data and linkage to a well-established demographic surveillance system, with few losses to follow-up. Limitations are lack of accurate gestational age estimation, unknown birth weight of most participants and that individual socioeconomic data were unavailable. We did not have clinical data collected at discharge, which may be of value in taking a risk-based approach to post-discharge care. This analysis is from a single hospital and excludes residents outside KHDSS who may have different exposures and risks.

### CONCLUSIONS

Neonatal and YI admissions account for an increasing proportion of inpatient paediatric admissions, and their overall mortality rate remains high. Post-discharge mortality accounts for a lower proportion of all deaths than hospital admissions aged 2–59 months but the post-discharge mortality rate among YI is higher.[31 33] This is likely because of the predominance of fatal neonatal conditions such as extreme prematurity or birth asphyxia. Services to address post-discharge mortality are needed and should focus on infants at higher risk.

**Acknowledgements** We thank the parents, patients and staff of Kilifi County Hospital and the KEMRI-Wellcome Trust Research Programme for their participation in the study. This study is published with the permission of the Director, KEMRI. This work was supported, in whole or in part, by the Bill & Melinda Gates Foundation (Grant Number INV-00791). Under the grant conditions of the Foundation, a Creative Commons Attribution 4.0 Generic License has already been assigned to the Author Accepted Manuscript version that might arise from this submission.

**Contributors** AT: Conceptualisation, investigation, methodology, formal analysis, writing—original draft, writing—review and editing; MN: Conceptualisation, methodology, data curation, formal analysis, visualisation, writing—original draft, writing—review and editing; CO: Conceptualisation, investigation, methodology, formal analysis, validation, writing—original draft, writing—review and editing; AN: Investigation, methodology, project administration, writing—review and editing; MKM: Conceptualisation, methodology, writing—review and editing; NM: Investigation, project administration, funding acquisition, resources, supervision, writing—review and editing; NO: Data curation, writing—review and editing; MO: Data curation, writing—review and editing; JB: Conceptualisation, investigation, methodology, funding acquisition, supervision, validation, writing—review and editing. AT and MN contributed equally to this paper. AT, the guarantor, accepts full responsibility for the finished work and the conduct of the study, had access to the data and controlled the decision to publish.

**Funding** Authors NM, AN, NO and MO, and staffing, facilities and resources were funded by the Wellcome Trust (203077_Z_16_Z). MKM was supported by a Wellcome Trust International Intermediate Fellowship (221997/Z/20/Z). JB was supported by the Medical Research Council–Department for International Development–Wellcome Trust Joint Global Health Trials scheme (MR/M007367/1). JB and MN were supported by the Bill & Melinda Gates Foundation (OPP1131320). CO was supported by the Drugs for Neglected Diseases initiative/Global Antibiotic Research and Development Partnership (OXF-DND02). AT was supported by Crosslinks.

**Competing interests** JB declares the following: Chair of the DSMB for 'Efficacy and safety of whole-body chlorhexidine cleansing in reducing bacterial skin colonisation of hospitalised neonates - a pilot trial'. St George's, University of London and global sites; Treasurer of the Commonwealth Society for Paediatric Gastroenterology & Nutrition.

**Patient and public involvement** Patients and/or the public were not involved in the design, or conduct, or reporting, or dissemination plans of this research.

**Patient consent for publication** Not applicable.

**Ethics approval** Ethics Committee: Kenya Medical Research Institute (KEMRI) National Ethics Review Committee, Reference no. SCC 2778. Participants gave informed consent to participate in the study before taking part.

**Provenance and peer review** Not commissioned; externally peer reviewed.

**Data availability statement** Data are available in a public, open access repository. Data are available in a public, open access repository. De-identified participant data and analysis code have been deposited and may be accessed at the Harvard Dataverse via this link https://doi.org/10.7910/DVN/0XJVFX.

**ORCID iDs**
Alison Talbert http://orcid.org/0000-0002-9328-6903
James Berkley http://orcid.org/0000-0002-1236-849X

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
