## [Reviewer comments · BMJ Open]

ARTICLE DETAILS

TITLE (PROVISIONAL)	Trends in inpatient and post-discharge mortality among young infants admitted to Kilifi County Hospital, Kenya, a retrospective cohort study.
AUTHORS	Talbert, Alison; Ngari, Moses; Obiero, Christina; Nyaguara, A.; Mwangome, MK; Mturi, Neema; Ouma, Nelson; Otiende, M.; Berkley, James

VERSION 1 – REVIEW

REVIEWER	To Strand University of Bergen
REVIEW RETURNED	21-Sep-2022

GENERAL COMMENTS	This manuscript describes infant mortality, causes, associated factors, and admission trends over ten years in Kilifi county in Kenya. The main points from this work are that the proportion of admitted young infants and pre-discharge young infant mortality increased throughout the ten-year study period, half of all infant deaths were at home, and half of the post-discharge deaths were within one month after discharge. The manuscript also includes interrupted time-series analyses (ITS) to estimate the effect of the introduction of free maternity care in 2013. In other words, the manuscript is mainly descriptive, and the discussion reflects the findings and ambitions of the analyses well. The manuscript is well written and structured, and the analysis approach is reasonable and transparent. There is, however, the potential for improvements. 1. Lines 162 onward (on the nb regression models). The description of the analyses should be improved. Is it necessary to mention that the models "included the dependent variable of interest"? , what is meant by the time variable? Is it the time denominator ("offset" or "exposure") or year of study?2. Please also describe the methods used to take repeated admissions into account (beyond "robust standard errors")3. A justification of the variable selection procedure described in lines 174 onwards would also be helpful. Why is different variable selection procedures used for the different outcomes /multiple models?4. The sample size calculation is based on a minimal effect size of an HR of 2.0. This is a rather large effect size; the method for calculating the sample size is not described, and the expected distribution of the chosen exposure variable (birth weight) is not described. Lastly, but probably most importantly, the sample size calculation does not appear to reflect the study's primary goals. The
---

	sample size is large, the main purpose is descriptive, and a single-exposure-based sample size calculation is not the right approach here. 5. The conclusion should reflect the results of this study only.
--	--

REVIEWER	Rachel Wangari Kimani The Rockefeller University, Neurobiology of language
REVIEW RETURNED	03-Oct-2022

GENERAL COMMENTS	Comments The manuscript is interesting and takes an interesting approach to analyze data on young infants aged less than 60 days. Abstract 1. Line 24 and 27. Clearly state that analysis in this paper is on only 4,421 KHDSS YI. 2. Page 7, line 99- sentence structure Results 3. Page 22, line 329- This sentence is unclear Discussion 4. Although there seem to be robust results, the discussion is scant. The authors intimated that the data would be vital for developing targeted interventions to improve survival, but the recommendations were vague, leaving the question, so what? For example - How do the results compare to national statistics, changes in demographics, and epidemiology in the area? Has the devolution of services influenced the trends? Most deaths were neonatal; is skilled birth attendance an issue? 5. Page 26 lines 380-381- implying that caregivers may be more likely to seek re-admission for YI or may live closer to KCH. What is the data on readmission that supports this statement? According to the results, there were only 142 readmissions out of 4272 6. Page 26-line 386 most YI but low admission weight <2.5kg was common (40%) - could you expand on this point since malnutrition is a pertinent issue? How does this contribute to reported deaths? 7. Line 388- Expand on "problems in the neonatal period" according to Table S4, 124 out of 208 post-discharge deaths were 0-2 days old. Is early discharge an issue? Is home birth an issue? Is there a delay in care? Socioeconomic background in the KHDSS area may be useful. 8. Line 388-What is the current meningitis treatment guidelines that are insufficient? What about other resources, such as the lack of mechanical ventilators? Limitations 9. Line 392- large sample size. However, over 7K were outside the surveillance system. This is a limitation of this study. The KHDSS area has more surveillance compared to other parts of the country- this affects the applicability of these findings. 10. Table S2 shows there is missing clinical information, including the severity of disease during admission. Did this affect the accuracy of the data? Conclusion: 11. Was the objective of the study achieved? What are the admission trends and inpatient and post-discharge mortality among YI between 2009-2019?
--

VERSION 1 – AUTHOR RESPONSE

Reviewer: 1

Dr. Tor Strand, University of Bergen

Comments to the Author:

This manuscript describes infant mortality, causes, associated factors, and admission trends over ten years in Kilifi county in Kenya. The main points from this work are that the proportion of admitted young infants and pre-discharge young infant mortality increased throughout the ten-year study period, half of all infant deaths were at home, and half of the post-discharge deaths were within one month after discharge. The manuscript also includes interrupted time-series analyses (ITS) to estimate the effect of the introduction of free maternity care in 2013. In other words, the manuscript is mainly descriptive, and the discussion reflects the findings and ambitions of the analyses well.

The manuscript is well written and structured, and the analysis approach is reasonable and transparent.

There is, however, the potential for improvements.

1. Lines 162 onward (on the nb regression models). The description of the analyses should be improved. Is it necessary to mention that the models "included the dependent variable of interest"? , what is meant by the time variable? Is it the time denominator ("offset" or "exposure") or year of study?

Thank you for this useful comment. We have edited the section to make it clearer. We have specified the time was time month from January 2009 to December 2019 and was used in the model as exposure to account for time since the beginning of the observations.

2. Please also describe the methods used to take repeated admissions into account (beyond "robust standard errors")

We had used the robust standard error approach that does not explicitly correct the measure of effect because the proportion of YIs with multiple admissions was very low: 3.5% and 3.4% YIs included in the post-discharge and inpatient analyses respectively had more than one hospital admission. To correct for patient clustering, we have now used the gamma distributed shared frailty Cox proportional hazards model. However, as we expected there was no significant change to the results and the final model AUCs have not changed. For the inpatient analysis, there are no validated competing risk models with either shared frailty or random effects, so we ran competing risk model with single admission per YI only (retaining the latest admission for those with multiple admission to ensure all inpatient deaths are captured). This model was not significantly different from the one with multiple admissions using robust standard errors as the changes did not alter the p-values and 95% CIs of individual independent variables. The AUC was not different from the one with multiple admissions: 0.88 (95%CI 0.86–0.90) versus 0.88 (95%CI 0.86–0.89). We therefore decided to retain the more complete model with multiple admissions. We have updated the statistical methods and results sections to reflect the changes. The new single admission competing risk multivariable model has been added to the supplementary tables Table S7.

3. A justification of the variable selection procedure described in lines 174 onwards would also be helpful. Why is different variable selection procedures used for the different outcomes /multiple models?

Both the inpatient and the post-discharge regression model used the backward stepwise approach. We understand the description in the post-discharge model could be misinterpreted, we have added the words "similar to the inpatient analysis" to make it clear the approaches used were similar.

4. The sample size calculation is based on a minimal effect size of an HR of 2.0. This is a rather large effect size; the method for calculating the sample size is not described, and the expected distribution of the chosen exposure variable (birth weight) is not described. Lastly, but probably most importantly,

the sample size calculation does not appear to reflect the study's primary goals. The sample size is large, the main purpose is descriptive, and a single-exposure-based sample size calculation is not the right approach here.

Thanks once more for this comment. This was a secondary analysis of routine hospital data linked to a community-based demographic health surveillance system. Therefore, we are not estimating the sample size needed for the study because all data for eligible YIs during the study period were included. We were cautious not to include the words 'post-hoc power' in the sample size statement because it is debatable whether it is necessary for a secondary analysis (<https://www.jrheum.org/content/jrheum/early/2022/05/10/jrheum.211115.full.pdf>). We have added a statement to the effect that all the eligible YIs were included in the analysis at line 202 page 12, therefore no sample size was estimated prior to undertaking the analysis.

5. The conclusion should reflect the results of this study only.

We have updated the conclusions both in the abstract and manuscript to reflect the study results only.

Abstract

"Conclusions

Neonatal and YI accounted for an increasing proportion of paediatric admissions and their overall mortality remains high. Post-discharge mortality accounts for a lower proportion of deaths but mortality rate is higher than among children aged 2-59 months. Services to address post-discharge mortality are needed and should focus on infants at higher risk."

Manuscript

"Conclusions

Neonatal and YI admissions account for an increasing proportion of inpatient paediatric admissions, and their overall mortality rate remains high. Post-discharge mortality accounts for a lower proportion of all deaths but the mortality rate is higher than among children aged 2 to 59 months. This is likely because of the predominance of fatal neonatal conditions such as extreme prematurity or birth asphyxia. Services to address post-discharge mortality are needed and should focus on infants at higher risk".

Reviewer: 2

Dr. Rachel Wangari Kimani, The Rockefeller University

Comments to the Author:

Comments

The manuscript is interesting and takes an interesting approach to analyze data on young infants aged less than 60 days.

Abstract

1. Line 24 and 27. Clearly state that analysis in this paper is on only 4,421 KHDSS YI.

This has been added:

***"Participants:* YI aged less than 60 days admitted to hospital January 2009 to December 2019: 12,271 admissions in 11,877 individuals. YI who were resident within a health and demographic surveillance system (KHDSS): n= 3,625 with 4,421 admissions were followed up for 1 year after discharge."**

***"Primary and secondary outcome measures:* Inpatient and 1 year post-discharge mortality, the latter in KHDSS residents."**

2. Page 7, line 99- sentence structure

Heading has been changed to "Study design and setting"

Results

3. Page 22, line 329- This sentence is unclear

This has been clarified as below:

"In the multivariable model, low birth weight, admission age <8 days, bacteraemia and clinical signs of severe illness were positively associated with inpatient mortality."

Discussion

4. Although there seem to be robust results, the discussion is scant. The authors intimated that the data would be vital for developing targeted interventions to improve survival, but the recommendations were vague, leaving the question, so what? For example - How do the results compare to national statistics, changes in demographics, and epidemiology in the area? Has the devolution of services influenced the trends? Most deaths were neonatal; is skilled birth attendance an issue?

There are very limited data on young infants: we could not find any national statistics or post-discharge statistics for this age group, besides a description of the neonatal inpatient deaths in 16 Kenyan hospitals. [Reference 5 Irimu G, Aluvaala J, Malla L, et al. Neonatal mortality in Kenyan hospitals: a multisite, retrospective, cohort study. BMJ Glob Health. 2021;6(5) doi 10.1136/bmjgh-2020-004475] In 2018/9 69% of births were reported to be attended by skilled health personnel in Kilifi County which is slightly higher than the national average (https://thinkwell.global/wp-content/uploads/2022/01/Kilifi-County-THS-UC-brief_20-December-2021.pdf)

We have now included the proportion of young infant admissions in our study who were born in KCH in table 1 of YI characteristics and in the univariate analysis (supplementary tables S3 and S4) and multivariate analysis (table 3) of risk factors for inpatient and postdischarge deaths. We do not have data on which of the outborn YIs were delivered at health facilities.

We have included the following on line 364 page 25 with references :

"Delivery by a skilled health worker has been shown to be effective in reducing perinatal mortality. We did not collect data on delivery by a skilled birth attendant but in 2018/9 69% of births in Kilifi County were reported to be attended by skilled health personnel which is slightly higher than the national average."

5. Page 26 lines 380-381- implying that caregivers may be more likely to seek re-admission for YI or may live closer to KCH. What is the data on readmission that supports this statement? According to the results, there were only 142 readmissions out of 4272

The 141 readmissions refer to those readmitted within the 1st 60 days of life (page 13 line 217). We described these for the purpose of the multiple admissions analysis. We did not set out to include readmissions during the 1 year follow up period as an outcome.

6. Page 26-line 386 most YI but low admission weight <2.5kg was common (40%) - could you expand on this point since malnutrition is a pertinent issue? How does this contribute to reported deaths?

In this age group it is difficult to distinguish low birth weight from malnutrition. We have reported the proportion of deaths in different admission weight categories in the supplementary materials. For example, 38% YIs admitted with weight <1.5kg died during inpatient phase, 15% among YIs with 1.5 to <2.5Kg and 8.5% among YIs with weight ≥2.5kg (Table S3). For the post-discharge deaths 8.5% YIs died among those with weight <1.5kg, 9.2% among 1.5 to <2.5kg and 3.7% among those with weight ≥2.5kg (Table S4).

We have included the following text on line 390 page 26:

"In young infants it is difficult to distinguish low birth weight from malnutrition, but we have reported the higher case fatality rates in the lower admission weight categories (Tables S3 and S4)"

7. Line 388- Expand on “problems in the neonatal period” according to Table S4, 124 out of 208 post-discharge deaths were 0-2 days old. Is early discharge an issue? Is home birth an issue? Is there a delay in care? Socioeconomic background in the KHDSS area may be useful.

Those aged less than 2 days at admission made up over half of the deaths in the 12 months’ post-discharge follow up period. Most were born in hospital and transferred to the paediatric ward from the postnatal ward. The analysis showed that longer rather than shorter admission durations were associated with post-discharge mortality (table 3) We agree that socioeconomic factors are likely important but we don’t have systematic data on that through the study period as it is not routinely collected in the KHDSS. We added a phrase that data on these were not available to line 398 page 27.

8. Line 388-What is the current meningitis treatment guidelines that are insufficient? What about other resources, such as the lack of mechanical ventilators?

The 2016 national guidelines state “If meningitis suspected and no LP performed: give IV/IM antibiotics for a minimum of 14 days. If Gram negative meningitis is suspected treatment should be IV for 3 weeks” . Mechanical ventilators were not available nor are they in the guidelines.

*We have added the following sentence on line page 25 line 371:
“Mechanical ventilation was not available in Kilifi County Hospital.”*

After we included the variable inborn (born at KCH) vs outborn in the multivariable analysis, the effect of bulging fontanel on post-discharge death was attenuated and laboratory features suggestive of meningitis were not associated so we deleted the sentence on bulging fontanel in the discussion.

Limitations

9. Line 392- large sample size. However, over 7K were outside the surveillance system. This is a limitation of this study. The KHDSS area has more surveillance compared to other parts of the country- this affects the applicability of these findings.

We agree that non-KHDSS residents could be different but follow up data were not available. We added this as a study limitation.

10. Table S2 shows there is missing clinical information, including the severity of disease during admission. Did this affect the accuracy of the data?

In all the analysis, the missing data were included as an extra category, so their effect was captured. Most clinical severity variables had less than 2% missing data except tachypnea (4.7%).

Conclusion:

11. Was the objective of the study achieved? What are the admission trends and inpatient and post-discharge mortality among YI between 2009-2019?

Yes. We reported the trend of increasing proportion of YI in under 5 admissions over the study period and the high inpatient mortality relative to post-discharge mortality.

Figures

12. Please provide captions for the figures on pages 36-38

Page 36: Figure 1. **Flow of study participants.**

Page 37: Figure 2. **Annual proportion of young infant admissions to all admissions <60 months, inpatient case fatality ratio (CFR) and post-discharge CFR.**

Proportions are plotted with 95% confidence intervals.

Page 38: Figure 3. **A: Monthly hospital admissions (with predicted mean temporal trend), B: Monthly case fatality rates (with predicted mean temporal trend), C: Monthly young infant**

inpatient case fatality before and after June 2013 and D: Monthly proportions of young infants to admissions <60 months old.